# Combining CAR T Cell Therapy and Oncolytic Virotherapy for Pediatric Solid Tumors: A Promising Option

Jiasen He [1], Faryal Munir [1], Dristhi Ragoonanan [1], Wafik Zaky [1], Sajad J Khazal [1], Priti Tewari [1], Juan Fueyo [2], Candelaria Gomez-Manzano [2] and Hong Jiang [2,*]

1   Department of Pediatrics—Patient Care, The University of Texas MD Anderson Cancer Center, Houston, TX 77030, USA
2   Department of Neuro-Oncology, The University of Texas MD Anderson Cancer Center, Houston, TX 77030, USA
*   Correspondence: hjiang@mdanderson.org

**Abstract:** Despite advances in treatment options, the clinical outcomes of pediatric patients with advanced solid tumors have hardly improved in decades, and alternative treatment options are urgently needed. Innovative therapies, such as chimeric antigen receptor (CAR) T cells and oncolytic viruses (OVs), are currently being evaluated in both adults and children with refractory solid tumors. Because pediatric solid tumors are remarkably diverse and biologically different from their adult counterparts, more research is required to develop effective treatment regimens for these patients. Here, we first summarize recent efforts and advances in treatments for pediatric solid tumors. Next, we briefly introduce the principles for CAR T cell therapy and oncolytic virotherapy and clinical trials thereof in pediatric patients. Finally, we discuss the basis for the potential benefits of combining the two approaches in pediatric patients with advanced solid tumors.

**Keywords:** CAR T cell therapy; oncolytic virus; oncolytic virotherapy; pediatric solid tumor; pediatric brain tumor; cancer immunotherapy

## 1. Introduction

Pediatric solid tumors have diverse pathophysiological characteristics and clinical presentations [1]. They can be classified as carcinomas, which are derived from epithelial cells, and as sarcomas, which are derived from mesenchymal cells. Unlike epithelial cancers that primarily manifest in adults, sarcoma is a more common type of solid tumor in pediatric populations [2,3]. It suggests differential mechanisms for tumor initiation and progression in these two age groups [3]. Based on Surveillance, Epidemiology and End Results data, approximately 60% of pediatric malignancies are solid tumors, including central nervous system (CNS) tumors (~20–23%), neuroblastoma (8–10%), Wilms tumors (7–8%), malignant bone tumors (such as osteosarcoma and Ewing sarcoma; ~7%), soft-tissue sarcomas (~7%), germ cell tumors (3–6%), liver tumors (hepatoblastoma and, more rarely, hepatocarcinoma; 0.5–2%), and retinoblastoma (2.5–3%) [4].

With advances in multidrug chemotherapy, surgery, and radiotherapy, the 5-year survival rate of all pediatric cancer patients now exceeds 80% [5]. These improvements, however, are mainly in patients with hematological malignances, while the prognosis for patients with solid tumors, particularly those with advanced disease, remains poor. Treatment-related toxicities of current therapies, along with limited therapeutic options for patients with relapsed and/or refractory diseases, are major challenges in the fight against pediatric cancer, and the need for novel treatment strategies is imperative [6]. Meanwhile, personalized targeted therapies and immunotherapies have been developed mostly in adults [5]. Although some of these novel therapies have been effective against certain cancers in adults, they have not been successfully and consistently reproduced

in children [7]. The failure may be attributed to that pediatric solid tumors are fundamentally distinct from adult solid tumors and usually have different embryonic origins and molecular and genetic profiles [2,3,7]. The Pediatric Cancer Genome Project showed that most pediatric cancers occur in developing mesodermal tissues, whereas most adult cancers occur in epithelial tissues [2,3]. Even if their histology is similar to that of adult cancers, pediatric cancers still have a remarkably different spectrum of mutations [2]. The first pediatric pan-cancer analysis identified 142 driver genes in pediatric cancer. Notably, only 45% of these genes were also identified in adult pan-cancer studies [8]. To this end, pediatric solid tumors mainly arise from the cells that acquire a deleterious mutation in genes that are both important for cell cycle arrest as well as organ differentiation during early organ development [3]. On the other hand, adult solid tumors originate within differentiated adult tissues with accumulation of multiple sequential mutations directly linked to environmental exposures [3].

Thus, it is not surprising that only a few targeted therapies are effective against pediatric solid tumors. For pediatric hematologic malignancies, ABL-class inhibitors and anti-CD antibodies are now universally adopted as a part of standard therapy, but for pediatric solid tumors, only a few drugs are under investigation in children, including larotrectinib for neurotrophic tyrosine receptor kinase (NTRK) fusion–positive tumors, crizotinib for inflammatory myofibroblastic tumors, MEK inhibitors for neurofibromatosis type 1, and BRAF inhibitors for BRAF V600E–mutated tumors [5,9,10].

Immunotherapy has shifted the paradigm of adult cancer treatment. It has been proven effective against a wide range of adult cancers, but this progress has not yet been fully translated into the pediatric cancer field. Immune checkpoint inhibitors, which are now widely used in adult anticancer regimens, tend to elicit less response in pediatric solid tumors with a low mutational burden [8,11,12]. To this end, in pediatric patients, immunotherapy is mainly limited to antibody-based therapies and CAR T cell therapy for hematologic malignancies. Dinutuximab, an antidisialoganglioside GD2 chimeric monoclonal antibody, is the only FDA-approved monoclonal antibody for pediatric solid tumors [13]. Although CAR T cell therapy has had dramatic effects against pediatric hematologic malignancies, its efficacy against pediatric solid tumors is currently suboptimal. The disparity of the effect of CAR T cell therapy in solid tumors is attributed to their unique properties that are different from hematologic malignancies, including physical barrier for CAR T cells to reach tumor cells, metabolically challenging and immunosuppressive tumor microenvironment (TME), and heterogeneous cancer cell populations within the tumors [14–16].

Oncolytic virotherapy is another alternative cancer treatment approach under active clinical investigation. The use of viruses for cancer treatment stems from the observations of tumor regressions that coincided with virus infection since the mid-1800s [17]. With the efforts of tuning the viruses to be more cancer-cell-selective, oncolytic virotherapy is emerging as a promising cancer treatment, especially in pediatric patients with brain tumors [18,19]. Preclinical evidence suggests that OVs, which can activate an TME, could be combined with adoptive cell therapy. However, because there is almost no clinical experience with this combination, the efficacy and toxicity are unknown.

## 2. CAR T Cell Therapy for Pediatric Solid Tumors

Adoptive cell therapy utilizes tumor-infiltrating lymphocytes as well as engineered CAR or T cell receptor-expressing T cells to target neoplastic cells [20]. CAR T cells recognize tumor-associated antigens (TAAs) on cancer cells' surface regardless of their expression of the major histocompatibility complex, leading to lysis of the targeted cancer cells [21]. This effect is achieved through a CAR that has an extracellular antigen-binding domain fused to a transmembrane domain and intracellular signaling elements from CD3$\zeta$, the initiator of T cell signaling [21]. Because an effective T cell response requires both T cell receptor signaling through CD3-$\zeta$ (signal 1) and costimulatory signaling (signal 2), the intracellular signaling elements of costimulators such as CD28, 4-1BB, and OX40 are inserted between

the transmembrane domain and the CD3ζ elements to improve and sustain the potency of the T cells [22] (Figure 1).

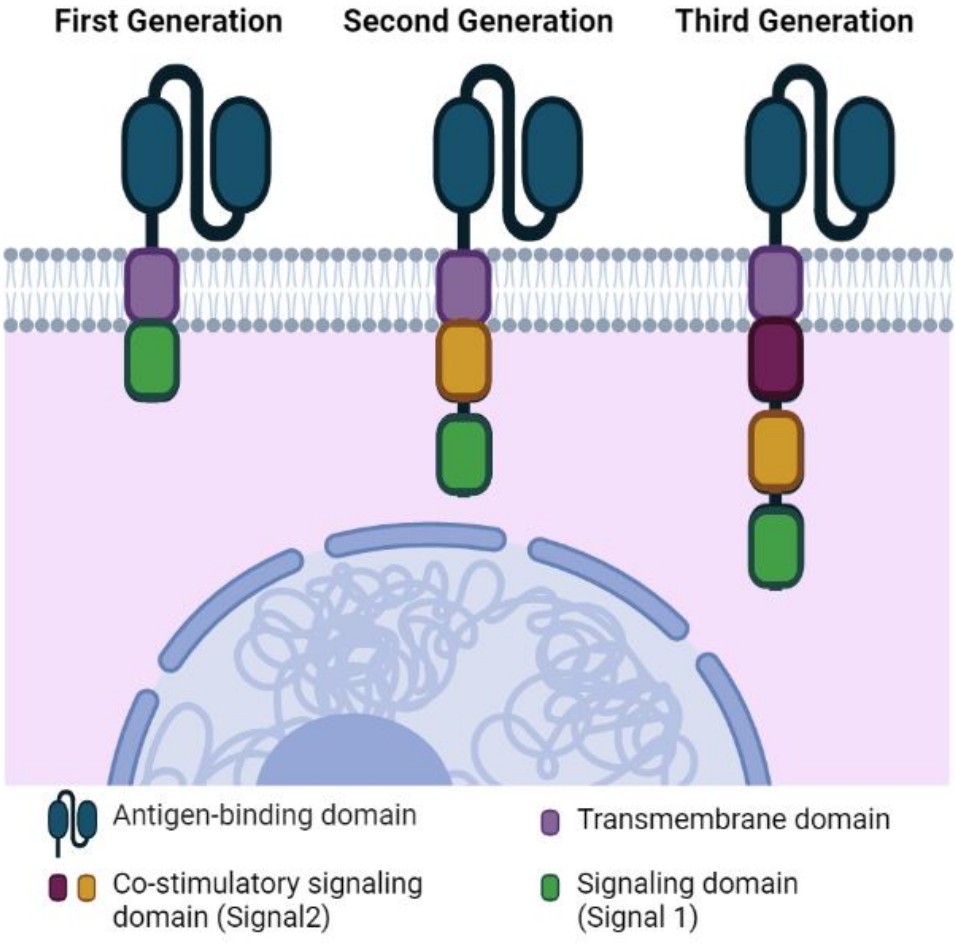

**Figure 1.** Evolving CAR designs in engineered T cells. The first-generation CARs contain an antigen-binding domain, transmembrane domain (TM), and a signaling domain (typically CD3ζ) that provides "signal 1" to activate T cells. A costimulatory signaling domain that provides "signal 2" is added in second-generation CARs, and two tandem costimulatory signaling domains are added in third-generation CARs between TM and signal 1 domain.

Thus far, the most successful CAR T therapy is targeting CD19, a surface protein that is exclusively expressed on both normal and malignant B cells [23]. With a reasonable safety profile, anti-CD19 CAR T cells are effective against B cell malignancies, including B cell acute lymphoblastic leukemia (B-ALL) and large B cell lymphoma (LBCL) [24,25]. After about three decades of research—going back to the development of the first generation of CAR T cells in the late 1980s [26,27]—anti-CD19 CAR T cells have become the first CAR T cell therapy approved by the U.S. Food and Drug Administration (FDA) for the treatment of patients with B cell malignances. The FDA approved Novartis's tisagenlecleucel for B-ALL in 2017, Gilead's axicabtagene ciloleucel for LBCL in 2017, Gilead's brexucabtagene autoleucel for mantle cell lymphoma in 2020, and Bristol Myers Squibb's lisocabtagene maraleucel for relapsed or refractory LBCL in 2021 [28]. To further improve patient outcomes, researchers are now focusing on identifying factors that affect the efficacy of the CAR T cells, such as disease histology, the lymphodepleting regimen used, and the CAR architecture/costimulatory domain employed in the construct [25].

Despite its remarkable success in patients with hematopoietic malignancies, however, the effect of CAR T cell therapy in treating solid tumors remains to be determined [14–16,25]. The application is challenging because of its innate limitations and the unique aforemen-

tioned properties of solid tumors [14–16]. Preclinical and clinical studies have identified several factors restraining the effective use of CAR T cells in patients with solid tumors. First, cancers treated with engineered T cells that target limited TAAs can undergo antigen escape owing to selection pressure favoring the tumor cells that lack the targeted antigens [29]. Second, because of limited available tumor-specific antigens, CAR T cells target TAAs that tumors may share with normal tissue, resulting in strong on-target, off-tumor toxicity [30]. Third, CAR T cells infiltrate solid tumors poorly and are prone to anergy within the immunosuppressive TME [31].

Clinical trials of CAR T cell therapies in pediatric patients with solid tumors are summarized in Table 1. Some of the ongoing trials are investigating CAR T cells targeting GD2, B7-H3 (CD267), human epidermal growth factor receptor 2 (HER2), New York esophageal squamous cell carcinoma 1 (NY-ESO-1), L1 cell adhesion molecule protein, epidermal growth factor receptor (EGFR), interleukin (IL)-13 receptor alpha 2 (IL13Rα2), glypican-3 (GPC-3), and prostate-specific membrane antigen (PSMA) [32,33]. Among these TAAs, GD2 is ranked twelfth among 75 potential targets for anticancer therapy by the National Cancer Institute [34]. It is overexpressed by a variety of pediatric and adult solid tumors, but its expression is limited in normal human tissues [32,33,35]. In a Phase I clinical trial investigating first-generation anti-GD2 CAR T cells without lymphodepletion, 3 of 11 patients with active neuroblastoma had a complete remission [36,37], which was associated with the persistence of the CAR T cells in the blood [37]. Thereafter, a Phase I trial demonstrated that third-generation anti-GD2 CAR T cells with lymphodepletion elicited modest early antitumor responses in patients with relapsed/refractory neuroblastoma [38] and that lymphodepletion with cyclophosphamide and fludarabine increased CAR T cell expansion [38]. Recently, a Phase I trial of anti-GD2 CAR T cells in patients with relapsed/refractory neuroblastoma showed that the CAR T cells did not persist until the dose was at least $1 \times 10^8$ cells/m$^2$ [39]. In another study, researchers transduced an anti-GD2 CAR gene into natural killer T (NKT) cells, an innate-like T cell sub-lineage possessing natural killer cell-like properties [40,41]. The resultant anti-GD2 CAR NKT cells, which were also engineered to express IL-15 to promote greater persistence [41], induced objective responses in patients with Stage IV relapsed/refractory neuroblastoma [40,41] (https://www.onclive.com/view/car-nkt-cell-therapy-can-induce-complete-remissions-in-pediatric-neuroblastoma (accessed on 1 December 2022)). Ongoing trials are investigating approaches that simultaneously target multiple proteins, such as the combination of CD276, PSMA, and GD2 (NCT04637503), to achieve sustainable tumor regression.

**Table 1.** Clinical trials of CAR T cell therapy in pediatric patients with solid tumors.

| Cancer Type | Phase | NCT | Age Range, Years | Cell Target | Route of Delivery for CAR T Cell | CAR T Cell Therapy | Cotherapy | Status |
|---|---|---|---|---|---|---|---|---|
| Solid tumors | I/II | NCT04432649 | 1–75 | B7-H3 (CD267) | Intravenous | Anti-CD267 4S CAR T cells | N/A | Recruiting |
| Relapsed/refractory non-CNS solid tumors | I | NCT04483778 | 0–26 | B7-H3 (CD267) | Intravenous | 4-1BBζ B7H3-EGFRt-DHFR; 4-1BBζ CD19-Her2tG | N/A | Recruiting |
| CD267-positive advanced solid tumors | I | NCT04864821 | 1–70 | B7-H3 (CD267) | Intravenous | Anti-CD267 CAR T cells | N/A | Not yet Recruiting |
| Relapsed/refractory CD267-positive solid tumors | I | NCT04897321 | 0–21 | B7-H3 (CD267) | Intravenous | Anti-CD267 CAR T cells | Lymphodepletion with cyclophosphamide and fludarabine | Recruiting |
| Solid tumors | N/A | NCT04691713 | 3–70 | B7-H3 (CD267) | Intravenous | Anti-CD267 CAR T cells | N/A | Recruiting |
| DIPG and relapse/refractory brain tumors | I | NCT04185038 | 1–26 | B7-H3 (CD267) | Locoregional | Anti-CD267 CAR T cells | N/A | Recruiting |
| Non-CNS solid tumors | I | NCT03618381 | 1–30 | EGFR | Intravenous | 4-1BBζ EGFR806-EGFRt; 4 1BBζ CD19-Her2tG | N/A | Recruiting |
| Relapsed/refractory brain tumors | I | NCT03638167 | 1–26 | EGFR806 | Locoregional | Anti EGFR806-specific CAR T cells | N/A | Recruiting |
| High risk and/or relapsed/refractory NB or other GD2-positive solid tumors | I/II | NCT03373097 [42] | 1–25 | GD2 | Intravenous | Anti-GD2 CAR T cells | N/A | Recruiting |
| Relapsed/refractory NB | I | NCT02761915 [39] | 0–1 | GD2 | Intravenous | Anti-GD2 CAR T cells | Lymphodepletion with leukapheresis, cyclophosphimide, fludarabine | Complete |
| GD2-positive OS, NB, or melanoma | I | NCT02107963 | 1–35 | GD2 | Intravenous | Anti-GD2 CAR T cells | Lymphodepletion cyclophosphamide and AP1903 | Complete |
| Relapsed/refractory solid tumors | I/II | NCT02992210 | 1–65 | GD2 | Intravenous | Anti-GD2 4S CAR T cells | N/A | Unknown |
| NB | I | NCT01822652 | All ages | GD2 | Intravenous | iC9-GD2 CAR T Cells | Lymphodepletion with Cyclophosphamide, fludarabine, pembrolizumab, other PD-1 inhibitors | Active not recruiting |
| NB | I | NCT00085930 | 1–21 | GD2 | Intravenous | Anti-GD2 CAR EBV-specific CTLs | N/A | Active not recruiting |
| DIPG or spinal DMG | I | NCT04196413 | 2–30 | GD2 | Intravenous | Anti-GD2 CAR T cells | Lymphodeption with fludarabineand cyclophosphamide | Recruiting |
| GD2-postive brain tumors | I | NCT04099797 | 1–21 | GD2 | Intravenous | C7R-GD2.CAR T cells | lymphodepletion chemotherapy | Recruiting |
| NB | I | NCT03294954 [40] | 1–21 | GD2 | Intravenous | Anti-GD2CAR NKT cells expressing IL-15 | Lymphodepletion with cyclophosphamide and fludarabine | Recruiting |
| Solid tumors | I/II | NCT05437315 | 1–75 | GD2 PSMA | Intravenous | Bi-4SCAR GD2/PSMA T cells | N/A | Recruiting |
| NB | I/II | NCT04637503 | 1–65 | GD2, CD276 (B7-H3), PSMA | Intravenous | Anti-GD2, PSMA, and CD276 CAR-T cells | N/A | Recruiting |
| GPC3-positive solid tumors | I | NCT04377932 | 1–21 | GPC3 | Intravenous | Anti-GPC3 CAR T cells | Lymphodeption with fludarabineand cyclophosphamide | Recruiting |
| GPC3-positive solid tumors | I | NCT04715191 | 1–21 | GPC3 | Intravenous | 15.21.GPC3-CAR T cells | Lymphodeption with fludarabineand cyclophosphamide | Not yet recruiting |
| Liver cancer | I | NCT02932956 | 1–21 | GPC3 | Intravenous | Anti-GPC3 CAR T cells | Lymphodeption with fludarabineand cyclophosphamide | Active not recruiting |
| Advanced sarcomas | I | NCT00902044 [43] | All ages | HER2 | Intravenous | Anti-HER2 CAR T cells | Lymphodeption with fludarabineand cyclophosphamide | Active not recruiting |
| Brain tumors | I | NCT03500991 [44] | 1–26 | HER2 | Locoregional | Anti-HER2 CAR T cells | N/A | Recruiting |

**Table 1.** *Cont.*

| Cancer Type | Phase | NCT | Age Range, Years | Cell Target | Route of Delivery for CAR T Cell | CAR T Cell Therapy | Cotherapy | Status |
|---|---|---|---|---|---|---|---|---|
| GBM | I | NCT01109095 | All ages | HER2 | Intravenous | Anti-HER 2 CAR CMV-specific CTLs | N/A | Complete |
| Relapsed/refractory Brain tumors | I | NCT02442297 | >3 | HER2 | Locoregional | Anti-HER2 CAR T cells | N/A | Recruiting |
| Relapsed/refractory IL13Rα2-positive malignant glioma | I | NCT02208362 [45] | 12–75 | IL13Rα2 | Locoregional | IL13(EQ)BBzeta/CD19t+ TCM-enriched T cells | N/A | Active |
| Relapsed/refractory IL13Rα2-positive malignant glioma | I | NCT04510051 | 4–25 | IL13Rα2 | intraventricularly | IL13(EQ)BBzeta/CD19t+ TCM-enriched T Cells | Lymphodepletion | Recruiting |
| Sarcoma, osteosarcoma, or Ewing sarcoma | I/II | NCT03356782 | 1–75 | Sarcoma cell surface antigens | IV | Sarcoma-specific CAR T cells | N/A | Recruiting |

Abbreviations: NB: neuroblastoma, OS: osteosarcoma, DIPG: diffuse midline intrinsic pontine glioma, DMG: diffuse midline glioma, GBM: glioblastoma, GD2: disialoganglioside, HER2: human epidermal growth factor receptor 2, EGFR: epidermal growth factor receptor, IL13Rα2: interleukin-13 receptor alpha 2, GPC3: glypican-3, PSMA: prostate-specific membrane antigen, CTL: cytotoxic T lymphocytes.

Another popular target for CAR T cell therapy is HER2. In one case report, a patient with colorectal cancer and lung metastasis received an intravenous infusion of anti-HER2 CAR T cells at a dose of $1 \times 10^{10}$ cells/m$^2$ and developed fatal respiratory failure 15 min later [46], possibly because of overwhelming cytokine release syndrome (CRS) resulting from the large dose of cells, which accumulated in the lung [46]. Then, in a Phase I/II trial in patients with recurrent/refractory HER2-positive sarcoma, anti-HER2 CAR T cells given at a lower starting dose of $1 \times 10^4$ cells/m$^2$ had no dose-limiting toxicities but also did not expand. When the dose was increased to more than $1 \times 10^6$ cells/m$^2$, the persistence of the CAR T cells was enhanced [47]. In an ongoing Phase I trial (NCT00902044), anti-HER2 CAR T cells given at a dose of up to $1 \times 10^8$ cells/m$^2$ with lymphodepletion elicited a clinical response in one patient with metastatic rhabdomyosarcoma [43].

CAR T cell therapy is also being investigated in pediatric brain cancers, which are the most common type of pediatric solid cancer and the leading cause of death from cancer in children [48]. Although the survival rates of children with medulloblastoma and low-grade glioma have improved remarkably, the prognosis for children with other brain tumors, such as diffuse midline glioma (DMG) and other high-grade gliomas, remains poor [49]. In 2016, the first case of effective CAR T cell therapy in a brain tumor patient was reported [45]. The patient, a 50-year-old man with recurrent multifocal glioblastoma, received multiple infusions of CAR T cells targeting IL13Rα2 over 220 days through the resected tumor cavity and then the ventricular system [45]. The patient had a transient complete response followed by relapse at various locations [45]. Thereafter, a Phase I study investigated the use of peripherally administered anti-HER2 CAR-modified virus-specific T cells (HER2-CAR VSTs) to treat progressive glioblastoma in pediatric and adult patients [50]. In addition to the signal from the HER2 antigen, the CAR T cells were stimulated by latent virus antigens presented by antigen-presenting cells (APCs) to optimize the T cells' persistence. Thus, HER2-CAR VSTs were detected in the peripheral blood up to 12 months after infusion [50]. Of the 17 patients enrolled in the study, one had a partial response for more than 9 months and seven had stable disease for up to 29 months [50]. Multiple factors may have contributed to the limited efficacy of the HER2-CAR VSTs in this trial. Given a previous report of a patient dying from HER2-CAR T cell therapy [46], the trial runners used a cautious dose-escalation strategy, which could have affected the efficacy. Although HER2-CAR VSTs remained in the peripheral blood for a long duration, whether they could cross the blood–brain barrier and be effective against the tumor was unknown [50]. Recently, another group presented their interim analysis of a study in which pediatric patients with recurrent/refractory CNS tumors received locoregional infusions of HER2-specific CAR T cells through CNS catheters (NCT0300991) [44]. The first three patients enrolled in this trial tolerated infusions into either the tumor cavity or ventricular system and experienced no dose-limiting toxicity. The researchers reported clinical and laboratory evidence of local CNS immune activation [44].

In early 2022, Majzner et al. reported encouraging results from a Phase I dose-escalation trial of anti-GD2 CAR T cell therapy in patients with H3K27M-mutated DMG [51], one of the most devastating pediatric tumors with an expected overall survival duration of around 12 months after RT [52]. The first four patients enrolled in this trial received intravenous or intracerebroventricular infusions of anti-GD2 CAR T cells and had manageable toxicity. Of these four patients, three showed clinical and radiological improvement. However, in addition to having CRS and immune-effector-cell-associated neurotoxicity syndrome (ICANS), which often occur with other CAR T cell therapies, some patients had tumor-inflammation-associated neurotoxicity, which was consistent with CAR T cell-mediated inflammation in sites of CNS disease and manifested as transient worsening of existing deficits or even as episodes of increased intracranial pressure due to brainstem edema. Because such neurotoxicity can be life-threatening, intensive inpatient management was required to ensure safety [51]. One of the major concerns in the trial was the therapy's on-target, off-tumor toxicity, since normal neural cells also express GD2. However, none of the patients showed any signs or symptoms of on-target, off-tumor toxicity, which

is consistent with the theory that CAR T cell therapy requires high antigen density for effector function [51]. Most recently, Vitanza et al. reported the preliminary results for the first three DMG patients who received intraventricular infusions of anti–B7-H3 CAR T cells in the ongoing Phase I BrainChild-03 trial [53]. Each patient received 40 infusions of anti-B7-H3 CAR T cells through a CNS catheter without lymphodepletion, and none had any dose-limiting toxicities. In addition, one patient sustained clinical and radiographic improvement [53]. These preliminary results demonstrate the feasibility of using repeated intraventricular infusions of CAR T cells without lymphodepletion.

Until now, only Phase I/II clinical trials with small patient numbers (mostly fewer than 20) have investigated CAR T cell therapy in pediatric patients with solid tumors. Although these trials have not yet produced conclusive results, they have provided proof-of-principle data supporting further clinical investigations. In most of the patients in these trials, CAR T cells lacked both toxicity and efficacy, but a small subset of patients had objective responses with limited on-target, off-tumor toxicity. Because the toxicity and efficacy are both dose-dependent and happen concomitantly, CAR T cell regimens need to be designed to have maximal efficacy and minimal toxicity through optimized dosing, delivery routes, and supportive care. To this end, the regional delivery of CAR T cells to CNS tumors induced objective responses [45,51,53] and may have diminished the toxicity related to upregulated cytokines in the peripheral blood [51]. Learning from clinical experience, researchers are placing greater efforts on identifying feasible tumor-specific targets; genetically engineering CAR T cells to increase and sustain their potency while minimizing their toxicity; and combining CAR T cells with other therapies to improve patient outcomes.

### 3. Oncolytic Virotherapy in Pediatric Solid Tumors

OVs are naturally occurring or genetically engineered replication-competent viruses that preferentially lyse cancer cells by selectively infecting and/or replicating in these cells [54–56]. OVs can be classified as DNA or RNA viruses based on their genomic content, which is packaged in a protein coat called the capsid [57,58]. In some OVs, the capsid is surrounded by lipid bilayer envelopes that are derived from portions of the host cell membrane, which includes some viral glycoproteins [58]. From 2000 to 2020, 97 studies reported data on 3233 patients enrolled in clinical trials of OVs, including adenovirus, herpesvirus, picornavirus, measles virus, vaccinia virus, reovirus, poliovirus, coxsackievirus, vesicular stomatitis virus (VSV), parvovirus, and retrovirus [59,60]. Thus far, only four OVs have been approved for the treatment of advanced solid tumors. Rigvir, a nonenveloped RNA virus derived from the native ECHO-7 strain of a picornavirus, was approved for melanoma treatment in Latvia in 2004 [61]. One year later, H101, a genetically modified adenovirus (DNA virus, nonenveloped), was approved in China for the treatment of nasopharyngeal carcinoma in combination with cytotoxic chemotherapy [62]. In 2015, the U.S. FDA approved talimogene laherparepvec, an attenuated herpes simplex virus, type 1 (HSV-1; DNA virus, enveloped) expressing granulocyte-macrophage colony-stimulating factor for the local treatment of unresectable cutaneous, subcutaneous, and nodal lesions in patients with recurrent melanoma after initial surgery [63]. Most recently, intertumoral G47 Δ, a third-generation oncolytic HSV-1, was approved in Japan for the treatment of recurrent glioblastoma [64]. However, among the more than 200 clinical trials of OVs, only 10 included pediatric patients [59,65], who accounted for 1.9% of all patients in the trials [59].

When interest in their use resurged in the 1990s, OVs were expected to cause a cascading oncolytic effect in the entire tumor, resulting in the eradication of the malignancy [56]. However, the clinical experience with OVs revealed that patient outcomes were not as ideal as what was observed in cultured cells or in nude mice with no immune components [55]. Objective responses were reported in 9% of patients in the clinical trials [59]. The low efficacy of OVs in patients is attributed to obstacles for translating oncolytic virotherapy into clinic, including viral delivery, spread, resistance, and antiviral immunity [66]. To enhance intratumoral viral spread, OVs have been armed with enzymes to degrade extracellular

matrix proteins, such as hyaluronidase and metalloproteinase [66]. To circumvent antiviral immunity, several approaches have been adopted, such as combining OVs with immuno-suppressive drugs; using low-seroprevalent OVs, molecular engineering of chimeric OVs, switching viral coat proteins; and delivering OVs with cellular vehicles [66]. Nevertheless, for the past two decades, accumulating evidence has shown that, although host antiviral immunity can disrupt the oncolytic cascade of the viruses, the direct lysis of cancer cells by OVs is followed by the induction of potent antitumor immunity, which is crucial to their efficacy [54–56]. To this end, viral infection and replication lead to tumor necrosis and the subsequent recruitment of immune cells to the tumor to elicit an innate immune response followed by adoptive immunity against cancer neoantigens [56]. To further activate the immunosuppressive TME and increase the antitumor immunity instigated by OVs in solid tumors, researchers are developing strategies to combine OVs with immune modulators such as cytokines, immune checkpoint inhibitors, or immune costimulators [56].

Clinical experience demonstrates that the effectiveness of immunomodulatory strategies depends on the presence of a baseline immune response and on the stimulation of pre-existing immunity [31,67]. In solid tumors, an immunogenic (or "inflamed" or "hot") TME includes tumor-infiltrating lymphocytes, possible genomic instability, a pre-existing antitumor immune response, and tumor-associated immune cells that express programmed death-ligand 1 (PD-L1), whereas a nonimmunogenic (or "noninflamed" or "cold") TME lacks these components [31,67]. Immunogenomic analyses have shown that the functional orientation of the TME has a prognostic role in adults with solid tumors. A systematic analysis of public RNAseq data from 408 pediatric patients with five types of extracranial tumors revealed that the five principal modules for immune traits were the same as those described previously in adults with these tumors [68]. The best overall survival was correlated with the cluster characterized by low enrichment of the wound-healing signature, high Th1 infiltration, and low Th2 infiltration [68]. Thus, high-risk refractory pediatric solid tumors, including most pediatric brain tumors, particularly aggressive subtypes such as DMG and medulloblastoma, are immune-cold, with high myeloid signatures and low T cell infiltration [69–71]. As mentioned previously, preclinical and clinical studies have demonstrated that OVs can induce immune activation within the TME, turning cold tumors into hot ones [18,19,72–74]. Thus, oncolytic virotherapy is a promising alternative approach for these patients.

The few clinical trials of OVs in pediatric patients with relapsed/refractory solid tumors, including CNS neoplasms, are summarized in Table 2. These trials have demonstrated that the viruses can elicit inflammation within the TME with acceptable safety profiles and encouraging clinical responses [18,19,65,75]. In a Phase I trial in children with relapsed/refractory neuroblastoma, rhabdomyosarcoma, or rare tumors with neuroendocrine features, Seneca Valley virus (NTX-010), an oncolytic RNA virus (family *Picornaviridae*), was well tolerated either alone or in combination with cyclophosphamide at the dose levels tested [76]. This trial was based on the results of a Phase I trial of NTX-010 in adult patients with small cell lung cancer or carcinoid tumors. However, Phase II trials showed that NTX-010 given after platinum-based chemotherapy was not beneficial for adult patients with small cell lung cancer [77]. In fact, the persistence of the virus in the blood was associated with a shorter progression-free survival duration [77]. In two other Phase I trials, Seprehvir (HSV1716), an oncolytic HSV-1, was delivered either intratumorally or intravenously in young patients with relapsed/refractory extracranial solid tumors [78,79]. The trials showed that the OV treatment with either delivery method was well tolerated but did not elicit objective responses [78,79]. Alternative delivery methods to increase the amount of OVs administered to patients, minimize toxicities, and avoid direct tumor injections have been investigated. In a Phase I trial that included nine pediatric patients with relapsed/refractory solid tumors, autologous mesenchymal stem cells were used as the vehicle to deliver Celyvir, an oncolytic adenovirus, to the tumor sites [75]. Although the OV caused only Grade 1 toxicities, it did not elicit any objective responses [75].

Moreover, the trial had a high screening failure rate (around 50%), as many patients had disease progression during the 6 weeks needed to manufacture Celyvir [75].

**Table 2.** Clinical trials of oncolytic virotherapy in pediatric patients with solid tumors.

| Cancer Type | Phase | NCT | Age Range, Years | Virus Name | Virus Type/Family | Route of Delivery | Cotherapy | Status |
|---|---|---|---|---|---|---|---|---|
| Treatment-naïve DIPG or DMG | I | NCT03178032 [18] | 1–18 | Adenovirus (DNX-2401) | *Adenoviridae* | Intratumoral injection | Neoadjuvant therapy | Complete |
| Refractory retinoblastoma | I | NCT03284268 | 1–12 | Adenovirus (VCN-01) | *Adenoviridae* | Intravitreal injection | Systemic intraarterial or intravitreal chemotherapy or radiotherapy | Recruiting |
| Brain tumors | I/II | NCT03330197 | 0–21 | Adenovirus (Ad-RTS-hIL-12) | *Adenoviridae* | Intratumoral injection | Oral Vekedimex | Terminated |
| Recurrent high-grade gliomas | II | NCT04482933 | 3–21 | HSV G207 | *Herpesviridae* | Intratumoral injection | Radiation | Not yet R |
| Recurrent cerebellar solid tumors | I | NCT03911388 | 3–18 | HSV G207 | *Herpesviridae* | Intratumoral injection | - | Recruiting |
| Recurrent CNS supratentorial neoplasms | I | NCT02457845 | 3–18 | HSV G207 | *Herpesviridae* | Intratumoral injection | Radiation | Active, not yet recruiting |
| Non-CNS solid tumors | I | NCT00931931 | 7–30 | HSV1716 | *Herpesviridae* | Intratumoral injection or intravenous | - | Complete |
| Recurrent childhood CNS solid tumors that can be removed by surgery | I | NCT02031965 | 12–21 | HSV-1716 | *Herpesviridae* | Intratumoral injection | Dexamethasone, conventional surgery/resection | Terminated |
| Recurrent MB or recurrent ATRT | | NCT02962167 | 1–39 | Modified Measles Virus (MV-NIS) | *Paramyxoviridae* | Intratumoral injection or intrathecal | - | Recruiting |
| GBM, NB, or sarcoma | I/II | NCT01174537 | 3–75 | NDV | *Paramyxoviridae* | Intravenous | - | Withdrawn |
| Metastatic cancers resistant to conventional anticancer treatments | II | NCT00348842 | All ages | NDV | *Paramyxoviridae* | Intratumoral injection or intravenous | - | Withdrawn |
| Recurrent malignant gliomas | Ib | NCT03043391 | 12–21 | Polio/Rhinovirus Recombinant; PVS-RIPO | *Picornaviridae* | Intratumoral injection | - | Active, not yet recruiting |
| Non-CNS solid tumors | I | NCT01169584 | 2–21 | Recombinant Vaccinia Virus | *Poxviridae* | Intratumoral injection | - | Complete |
| Non-CNS bone and soft tissue sarcomas metastatic to the lung | II | NCT00503295 | >16 | Reovirus (REOLYSIN®) | *Reoviridae* | Intravenous | - | Complete |
| Relapsed/refractory ST with neuroendocrine features | I | NCT01048892 | 3–21 | Seneca Valley virus-001 | *Picoranviridea* | intravenous | Cyclophosphamide | Complete |

Abbreviations: NB: Neuroblastoma, OS: Osteosarcoma, MB: Medulloblastoma, ATRT: Atypical teratoid rhabdoid tumor, DIPG: Diffuse midline intrinsic pontine glioma, DMG: Diffuse midline glioma, GBM: Glioblastoma, ST: Solid tumor, CNS: Central nervous system, GBM: Glioblastoma; ST: Solid tumor, HSV: Herpes simplex virus, NDV: Newcastle disease.

Two recent Phase I trials of OVs in pediatric patients with high-risk cold CNS tumors had some encouraging results. In one trial, 12 pediatric patients with recurrent or progressive supratentorial high-grade glioma each received a single intratumoral injection of G207, an oncolytic HSV-1 [19]. Of these 12 patients, 11 had responses, and the median overall survival duration was 12.2 months [19]. Most treatment-related toxicities were Grade 1, and no treatment-related Grade 3 or 4 toxicities were reported [19]. In addition, six of the patients had received radiotherapy as part of their standard treatment [19]. In the other trial, 12 pediatric patients with DMG each received a single intratumoral injection of DNX-2401 (Delta-24-RGD), an oncolytic adenovirus that selectively replicates in tumor cells with an aberrant Rb/E2F pathway, followed by radiotherapy [80]. The median survival duration of the patients was 17.8 months, and most of them had only Grade 1 or 2 adverse events [18]. In both trials, the viruses markedly increased the number of tumor-infiltrating lymphocytes [18,52], indicating that they can convert cold tumors into hot tumors to instigate an adoptive immune response.

In summary, as in the aforementioned clinical trials of CAR T cell therapy, most clinical trials of OVs in pediatric patients with solid tumors are small, often enrolling fewer than 20 patients. Although most trials did not show objective responses, the recent two trials in patients with CNS tumors suggest that OVs are effective against immunosuppressive cold tumors that are resistant to conventional therapies [18,52]. Thus, the results are encouraging, albeit inconclusive. Moreover, the OVs tested thus far have not had unacceptable toxicity. These data justify further clinical investigations of OVs in this type of patients. In addition, as the two trials in patients with CNS tumors suggest, it is feasible to combine OVs with radiotherapy. Along these same lines, the OV-mediated immune activation in the TME may potentiate other immunotherapies that are dampened by an immunosuppressive TME, such as CAR T cells.

## 4. Combination of CAR T Cell Therapy and Oncolytic Virotherapy in Pediatric Solid Tumors

The suboptimal efficacy of CAR T cells in solid tumors can be attributed to the poor infiltration and inactivation of the T cells in the immunosuppressive TME [31,81]. Moreover, owing to the limited available targets for CAR T cells and the heterogeneity of solid tumors, the therapy may spare cancer cells with loss or downregulation of the target antigens, leading to tumor relapse [25,82,83]. However, OVs can activate the immunosuppressive TME, increasing the presence of T cells at the virus-injected tumor [18,19,72–74], and this immune activation can extend to disseminated untreated tumors and peripheral lymphoid organs [74,84]. Moreover, OVs can be armed with immune modulators, including proinflammatory cytokines, immune costimulators, or immune checkpoint inhibitors, to enhance their ability to activate the TME [56]. Meanwhile, the impaired APCs in the TME, such as dendritic cells (DCs) and tumor-associated macrophages (TAM), can be overturned and primed by TAAs in the cell debris from OV-mediated oncolysis [85,86]. Interferon gamma (IFNγ) induced by viral infection upregulates major histocompatibility complex (MHC) I expression on tumor cells [72,87], indicating that tumor cells can effectively function as APCs, especially if they are infected with a virus expressing an immune costimulator [73,88]. Thus, one can reasonably speculate that OVs can promote immunity against other TAAs in addition to the limited targets of CAR T cells, mitigating the cancer relapse encountered by CAR T cell therapy due to antigen escape. Moreover, CAR T cells, because they have instant potency, can fill the therapeutic gap that arises during the early stage of virotherapy, before the viruses are able to induce effective antitumor immunity. Collectively, these complementary properties make the two approaches a promising match for combination therapy (Figure 2).

Almost all published studies of combining CAR T cell therapy and OVs are preclinical studies (Table 3). These studies demonstrated that combining OVs with CAR T cells had therapeutic benefits in mice, including reduced tumor growth and metastasis and improved survival rates. Most of the studies were conducted in immunodeficient mice with human CAR T cells. Some of these studies investigated OVs that express chemokines (e.g., RANTES, CXCL11) or cytokines (e.g., IL-12, IL-15, tumor necrosis factor alpha (TNFα), IL-2) to enhance virus-mediated immune activation or increase the persistence of CAR T cells [89–96]. In other studies, researchers tried to increase efficacy by using OVs that express other transgenes, including mini-antibodies to block immune checkpoint receptors from binding with their partner ligands [90,93,96–98]; CAR targets to enable effective cell therapy [99–101]; and bispecific T cell engagers to redirect CAR T cells towards other TAAs in the absence of CAR T cell targets to address tumor heterogeneity [93,96,102]. In a few of the studies, immunocompetent mice with syngeneic tumors were used to test the efficacy of the combination of mouse CAR T cells and OVs [99,101,103,104]. In these mice, even without lymphodepletion, the CAR T cell therapy and combination therapy show therapeutic benefit [101,103,104]. In most studies, OVs were injected intratumorally to attract intravenously administered CAR T cells to the virus-activated TME to increase oncolysis. In two studies using immunocompetent syngeneic tumor models, CAR T cells

were administered intratumorally simultaneously with or after OV injection [99,101]. In one recent study, the intravenous injection of CAR T cells loaded with an OV (VSV or reovirus) followed by intravenous OV delivery prolonged the survival of mice with subcutaneous melanoma or intracranial glioma tumors [104].

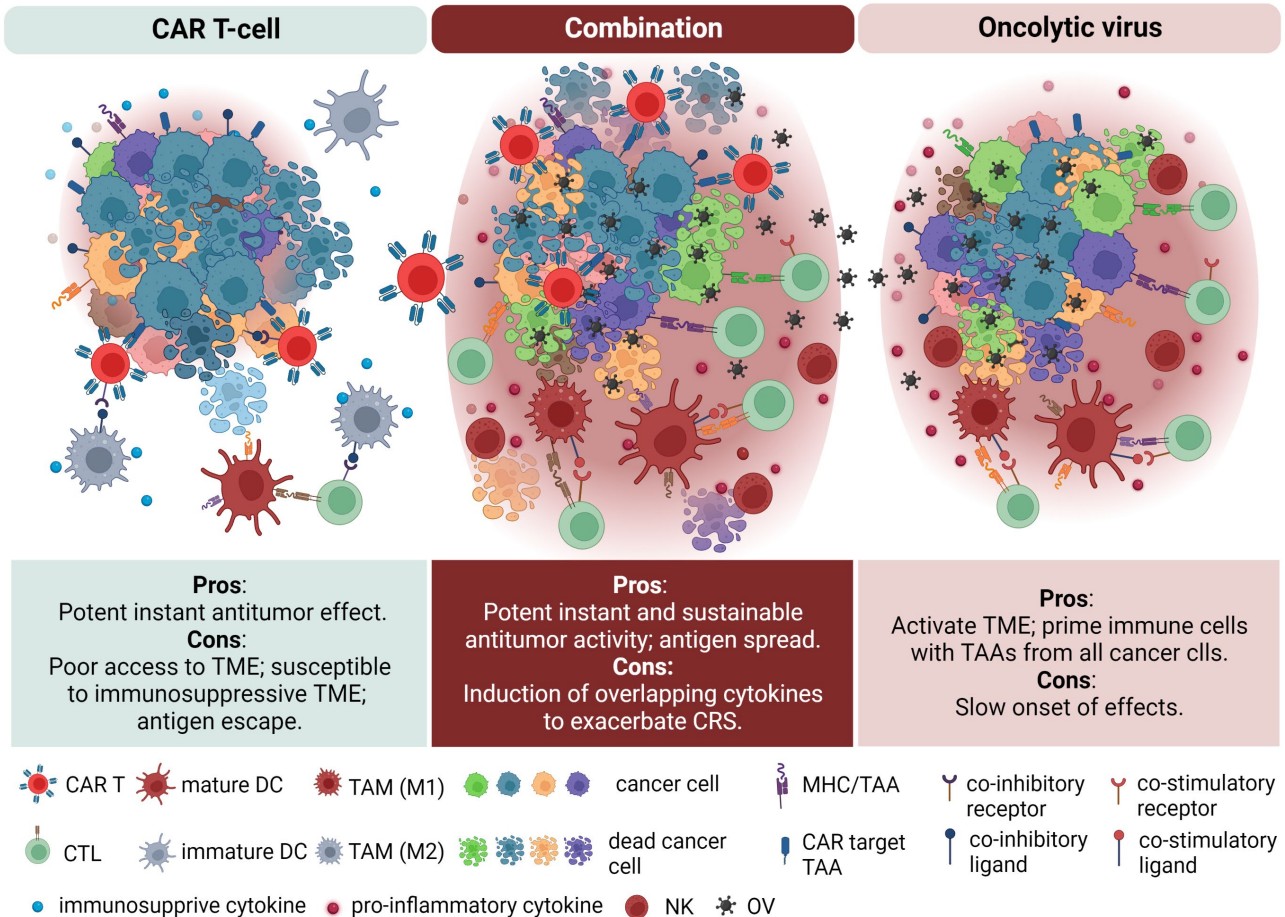

**Figure 2.** The effect of combining CAR T cell therapy and oncolytic virotherapy in tumor microenvironment (TME). CAR T cells have instant potent activity to kill cancer cells expressing tumor-associated antigens (TAAs) on cell surface, but are susceptible to TME with immune suppressive modulators, such as immunosuppressive cytokines (e.g., IL-10, TGFβ), immune checkpoint coinhibitory receptors and ligands (e.g., PD-1, PD-L1, CTLA-4), M2 phenotype tumor-associated macrophages (TAMs), etc. Oncolytic viruses (OVs) remodel the TME through upregulating proinflammatory cytokines (e.g., IFNγ, IL-6, TNFα, IL-12), immune checkpoint costimulatory receptors and ligands (e.g., OX40, OX40L, 4-1BB, 4-1BBL), mature dendritic cells (DCs), nature killer (NK) cells, M1 phenotype TAMs, etc. The oncolysis and immune activation mediated by OVs promote antigen spread, resulting in proliferation of cytotoxic lymphocytes (CTLs) targeting other TAAs presented by major histocompatibility complex (MHC) in addition to CAR T cells. The combination therapy takes advantage of the instant potent activity of CAR T cells and immune activation by OVs, leading to more effective lysis of the heterogeneous cancer cell populations to mitigate tumor relapse encountered by CAR T cell therapy due to antigen escape.

**Table 3.** Preclinical studies of combined CAR T cell therapy and oncolytic viral therapy in pediatric solid tumors.

| Cancer Type | Study Year/Author | CAR T Cell Target | Oncolytic Agent | Route of Delivery |
|---|---|---|---|---|
| Neuroblastoma | 2014/Nishio [89] | GD2 | Onc.Ad-Rantes/IL-15 | Intravenous CART. Intertumoral OAdV |
| Lung cancer | 2014/Wang [105] | HER2 | EphA2-TEA-VV | N/A |
| Breast or liver tumor | 2016/Slaney [106] | HER2, melanocyte protein (gp100) | VV-gp100 | Intravenous |
| Head and neck squamous cell carcinoma | 2017/Rosewell [90] | HER2 | CAdVECIL12p70/aPDL1 | Intravenous CART. Intertumoral CAdV |
| Prostate cancer or squamous cell carcinoma | 2017/Tanoue [97] | HER2 | CAdVEC-aPDL1 | Intravenous CART. Intertumoral CAdV |
| Pancreatic ductal carcinoma or colorectal carcinoma | 2018/Wing [102] | Folate receptor alpha | Onc.Ad-EGFR BiTE | Intravenous CART. Intertumoral OAd-BiTE |
| Pancreatic ductal carcinoma | 2018/Watanabe [103] | Mesothelin | Onc.Ad-TNFa/IL-2 | Intravenous CART. Intertumoral/Intravenous OV |
| Lung cancer | 2018/Moon [92] | Mesothelin | VV.CXCL-11 | Intravenous CART. Intertumoral/Intravenous OV |
| Breast cancer | 2019/Park [101] | CD19 | OV19t | Intravenous CART. Intertumoral OV19t |
| PDAC or squamous cell carcinoma | 2020/Porter [93] | HER2 | CAdTrio | Intravenous CART, Intertumoral CAdTrio |
| Breast cancer | 2020/Li [94] | Mesothelin | rAd.sT | N/A |
| Melanoma | 2020/Aalipour [99] | CD19 | mCD19VV | Intertumoral |
| Liver cancer or hepatocellular carcinoma | 2020/Tang [100] | CD19 | AdC68-TMC-tCD19 | N/A |
| B cell lymphoma | 2021/Wenthe [107] | CD19 | LOAd703 | Intravenous CART. Intertumoral LOAd703 |
| PDAC | 2021/Rosewell [96] | HER2 | CAdTrio | Intravenous CART, Intertumoral CAdTrio |
| GBM | 2021/Huang [98] | B7H3 | oAD-IL7 | Intravenous CART, Intertumoral oAD-IL7 |
| Solid tumor | 2021/Chen [95] | CD19 | rTTVΔTK-IL21 | Intravenous CART, Intertumoral rTTVΔTK-IL21 |
| Subcutaneous melanoma or intracranial glioma tumor | 2022/Evgin [104] | EGFRvIII | VSIV-mIFN β | Intravenous |
| GBM | 2022/Zhu [108] | CD70 | oHSV-1 | Intertumoral |
| GBM | 2022/Chalise [109] | LpMab-2 | G47 Δ (third-generation oncolytic HSV-1) | Intravenous CART. Intertumoral G45 Δ |

Abbreviations: GD2: disialoganglioside, HER2: human epidermal growth factor receptor 2, EGFR: epidermal growth factor receptor, PDAC: pancreatic adenocarcinoma, GBM: glioblastoma.

The combination of CAR T cell therapy and an OV is being investigated in only one clinical trial, in which HER2-CAR VSTs and a binary oncolytic adenovirus are being used to treat patients with advanced HER2-positive solid tumors (NCT03740256). The trial, whose enrollment period is from 14 December 2020 to 30 December 2024, will ultimately include about 45 patients who are age 18 years or older. The OV used in the trial, CAdVEC, is genetically modified to express currently undisclosed immunomodulatory molecules that may enhance the antitumor effects of endogenous T lymphocytes as well as those of adoptively transferred CAR T cells. The trial is sponsored by Baylor College of Medicine, which has already sponsored two Phase I clinical trials of anti-HER2 CAR T cell therapy in patients with glioblastoma (NCT01109095) and advanced solid tumors (NCT00902044) [43,47,50]. Both trials included pediatric patients. In the trial in glioblastoma patients, the researchers used HER2-CAR VSTs [50]. The two trials demonstrate that the infusion of HAR2 CAR T cells is safe and associated with clinical benefit, indicating that further evaluation of these cells in combination with OVs is warranted [43,47,50]. In addition, the Baylor researchers reported encouraging results from preclinical studies in which HER2 CAR T cells were combined with oncolytic adenoviruses armed with IL-12, an anti–PD-L1 antibody, and a bispecific T cell engager molecule specific for CD44 variant 6 [93,97].

## 5. Future Perspectives and Conclusions

On the basis of numerous preclinical and clinical investigations of CAR T cell therapy and oncolytic virotherapy, one can reasonably expect that combinations of these two interventions would have better efficacy than either intervention alone in cancer patients. Compared with adult cancers, pediatric cancers are relatively rare [110], and clinical trials in pediatric cancer patients tend to have low participant numbers. Thus, the findings of trials in adult patients with solid tumors are indispensable references for optimizing the antitumor effects of CAR T cell therapy, oncolytic virotherapy, and their combination in pediatric patients with solid tumors.

Potential long-term side effects have a much stronger impact on children than adults. For example, the random integration of the CAR gene into the genome through retro- or lentiviral vectors increases the risk of tumor development resulting from insertional mutagenesis [111,112]. Moreover, CAR T cells can result in significant on-target off-tumor toxicities given lack of identified exclusive targets in tumors [113]. Autoimmunity is also a concern in oncolytic virotherapy, albeit to a lesser extent, since OVs can promote the cross-priming of antigens in the tumor that are also expressed by normal cells [114,115].

Currently, the most common toxicities of CAR T cell therapy are CRS and ICANS, which pose challenges in its widespread use in the outpatient setting [113]. ICANS can occur concurrently with CRS or in the absence of CRS [113,116]. The mechanisms of CRS and ICANS are becoming clearer, but many aspects remain unknown. Patients with CRS whose symptoms include fever, tachycardia, and hypotension have increased levels of cytokines in their serum, including IL-6, IFN$\gamma$, IL-2, IL-2–receptor-$\alpha$, IL-8, and IL-10 [113]. Among these cytokines, IFN$\gamma$ and IL-6 play central roles in the innate immune response to bacterial and viral infections, which is followed by the upregulation of proinflammatory Th1 cells, chemokines, and other cytokines, such as TNF$\alpha$, IL-12, and IL-2 [117–120]. Moreover, some OVs are engineered to express proinflammatory cytokines such as IL-12, IL-2, and TNF$\alpha$. Therefore, more studies are needed to design combination treatment regimens whose additive or synergistic effects do not exacerbate the cytokine storm. For example, the localized delivery of fewer CAR T cells may achieve an effect equivalent to that of a higher dose of intravenously delivered CAR T cells but without inducing high cytokine levels in the peripheral blood [51]. In addition, instead of OVs expressing IL-12, IL-2 or TNF$\alpha$, OVs expressing IL-15 or immune costimulators, such as OX40L or inducible costimulator (ICOS), may be more appropriate for combining with CAR T cells [73,121]. Nevertheless, although OVs induce immune activation in the TME, not all virus-mediated effects favor CAR T cell activity. During the early stages of viral infection, the virus-mediated upregulation of type I IFNs promotes the acute apoptosis of memory T cells [122–124]. For example, an oncolytic VSV expressing IFN beta (IFN$\beta$) promoted significant CAR T cell attrition in an IFN$\beta$-dependent manner; in contrast, an oncolytic reovirus induced only moderate CAR T cell attrition [125]. Furthermore, oncolytic VSV, vaccinia virus, and Newcastle disease virus induce vasculature disruption to enhance tumor destruction [126–128], but this may limit the ability of intravenously delivered CAR T cells to reach the targeted tumor cells. Thus, one way to circumvent the OV-mediated conditions that are hostile to CAR T cells may be to first treat the tumor with CAR T cells before using OVs and deliver the viruses intratumorally.

In summary, more research is needed to optimize combinations of CAR T cells and OVs, including their dosage, delivery, and schedule, to maximize their efficacy while minimizing toxicity. By continuing to advance these two sophisticated interventions individually, we will have enriched knowledge and more choices for combinations to develop better therapeutic options to benefit pediatric patients with advanced refractory solid tumors.

**Author Contributions:** Conceptualization, J.H., H.J. and J.F.; validation, J.H., H.J. and F.M.; writing—original draft preparation, J.H. and H.J.; writing—review and editing, H.J., W.Z., F.M., D.R., S.J.K., P.T., J.F. and C.G.-M.; visualization, J.H. and H.J.; supervision, H.J., J.F. and C.G.-M.;

funding acquisition: H.J., J.F. and C.G.-M. All authors have read and agreed to the published version of the manuscript.

**Funding:** This work was supported by the National Institutes of Health (NIH) R01CA256006, P50CA127001, Chance for Life Foundation and Cure Starts Now-DMG/DIPG Collaborative, the University Cancer Foundation via the Institutional Research Grant program at The University of Texas MD Anderson Cancer Center. The funding bodies were not involved in the decision to publish or the preparation of the manuscript.

**Informed Consent Statement:** Not applicable.

**Data Availability Statement:** Not applicable.

**Acknowledgments:** We thank Joseph A. Munch, Scientific Editors in the Research Medical Library at The University of Texas MD Anderson Cancer Center, for editing this article.

**Conflicts of Interest:** The authors declare no conflict of interest.

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
