# Peer review of "Combining CAR T Cell Therapy and Oncolytic Virotherapy for Pediatric Solid Tumors: A Promising Option"

_2673-5601, doi:10.3390/immuno3010004_

Round 1

Reviewer 1 Report

This review article by He and colleagues explores the opportunity of combining oncolytic virotherapy (OV) with CAR T cell therapy in pediatric solid tumors.

The article is well written and illustrates the most recent advances in the field. It could be a useful reading for tumor immunologists and contains an exhaustive reference list to current and completed clinical trials. Nevertheless, some minor changes and additions could improve the quality of this article

-        in the first section, the authors underline how CAR T therapy is highly efficient on pediatric hematological malignancies, while it works poorly on solid tumors. The authors should speculate on this observation and underline some key aspects of the two disease environments

-        section 2 describes in detail the design of CAR T against pediatric solid tumors. A cartoon depicting the structure of CAR T cells could help the reader to better follow the description of the chimeric antigen receptor

-        in many sections of the paper (especially section 2, 3 and in the discussion) the authors illustrate some general mechanisms of toxicity of CAR T and of OV. The authors should consider coordinating this information in a dedicated section (which could be titled “adverse events of CAR T and OV” or something similar) and use it as a reference in the final discussion

-        a long paragraph in section 2 describes the limitations of CAR T therapy in the context of solid tumors. Although some references are correctly cited, others citations on the topic could be added. For example

o   Martinez and Moon - CAR T Cells for Solid Tumors -  Front Immunol 2019

o   Hou et al – Navigating CAR T cells through the solid TME – Nat Reviews Drug Discovery 2021

o   Boccalatte et al – Advances and hurdles in CAR T cell immune therapy for solid tumors – Cancers 2022

-        On line 57 “GD2” is written after the references but it has no corresponding sentence

-        On line 119, “New York esophageal squamous carcinoma 1” should be followed by its acronym

-        The legends of table 1 and figure 1 should be written in a different typographical layout and separated from the main text

Author Response

We appreciate the reviewer's comments. The point-by-point answers to the critiques are as follows.

-        in the first section, the authors underline how CAR T therapy is highly efficient on pediatric hematological malignancies, while it works poorly on solid tumors. The authors should speculate on this observation and underline some key aspects of the two disease environments

A: We added the information as advised (lines 75-79).

-        section 2 describes in detail the design of CAR T against pediatric solid tumors. A cartoon depicting the structure of CAR T cells could help the reader to better follow the description of the chimeric antigen receptor

A: We added Figure 1 to depict the CAR structures.

-        in many sections of the paper (especially section 2, 3 and in the discussion) the authors illustrate some general mechanisms of toxicity of CAR T and of OV. The authors should consider coordinating this information in a dedicated section (which could be titled “adverse events of CAR T and OV” or something similar) and use it as a reference in the final discussion

A: Adverse events in the clinical trials are caused not only by the therapy itself but also affected by the design of the trials. We have comments on the toxicities in some individual trials. For the toxicities of these two therapies that will affect the combination, we have discussed in Section 5.   

-        a long paragraph in section 2 describes the limitations of CAR T therapy in the context of solid tumors. Although some references are correctly cited, others citations on the topic could be added. For example

o   Martinez and Moon - CAR T Cells for Solid Tumors -  Front Immunol 2019

o   Hou et al – Navigating CAR T cells through the solid TME – Nat Reviews Drug Discovery 2021

o   Boccalatte et al – Advances and hurdles in CAR T cell immune therapy for solid tumors – Cancers 2022

A: We added the references as suggested (line 79, 135)

-        On line 57 “GD2” is written after the references but it has no corresponding sentence

A: We deleted the typo.

-        On line 119, “New York esophageal squamous carcinoma 1” should be followed by its acronym

A: We added the acronym (line 147).

-        The legends of table 1 and figure 1 should be written in a different typographical layout and separated from the main text

A: We used different font for the legends in the revised version. The tables and figures are inserted into the main text according to the instructions for authors from the journal.

Reviewer 2 Report

The manuscript summarized current progress on CAR T cell therapy and oncolytic virotherapy for pediatric solid tumors and as a combination therapy for pediatric solid tumors. The authors pointed out that CAR T cell therapy and oncolytic virotherapy can complement each other and synergize the anti-tumor cytotoxicity for pediatric solid tumors.

In general, the manuscript covered the topics of pediatric solid tumors, current CAR T cell therapy and oncolytic virotherapy field very well. In the introduction part, the authors mentioned that in contrast to in adults, sarcoma is a more common type of solid tumors in pediatric populations (line 26). However, later in the review, the authors did not elaborate much this difference between adult solid tumors and pediatric solid tumors in oncolytic virotherapy part and only mentioned this in CAR T cell therapy part. Such differences between adult solid tumors and pediatric solid tumors can be discussed more in the manuscript and what kind of changes are needed compared to adult solid tumors because of those differences. Another weak point of the review is, the authors tried to point out the combination of CAR T and oncolytic virus can be a promising solution, but the authors did not compare the combination with current other therapies other than CAR T and oncolytic virus alone. The superiority of the combination over other therapy strategies like immune checkpoint inhibition, multidrug chemotherapy, and radiotherapy is not thoroughly discussed. Furthermore, the auther can discuss some of the latest progress in CAR T cells like conditional CAR T and their benefit to pediatric solid tumors.

The language of the manuscript is professional and well-written. But there are still some small mistakes. For example, in line 121 to 122, the abbreviation of glypican-3 was still written glypican-3, but later in the caption of table 1 (line 145) it was written GPC-3. Such abbreviations need to be synchronized. Meanwhile, there are some small format problems that made the manuscript hard to read. There are lack of space lines between captions and main text (line 309-310 and line 375-376).

In summary, this manuscript is a very good reference for the onco-immunology community. The review is clear, comprehensive and of relevance to the field. The review summarized the latest progress in CAR T cells and oncolytic virotherapy for pediatric solid tumors. The statements and conclusions are drawn fairly and clearly.

Author Response

We appreciate the reviewer's comments and suggestions.

As suggested by the reviewer, we added more information on the difference between adult and pediatric solid tumors in Section 1. In this section, we also briefly discuss the inefficiency of immune checkpoint inhibition, chemotherapy and radiotherapy in pediatric solid tumors. Since CAR T cell therapy and oncolytic virotherapy are two novel therapies under investigation in clinic, our focus is on these two approaches. Because they demonstrate complimentary properties in treating solid tumors, our focus here is mainly on the potential benefit to combine these two therapies. In Table 2, some of the OV clinical trials include pediatric patients with sarcomas. Since the number of pediatric patients are much lower than adult patients and most trials did not show objective responses, it’s hard to make a clear conclusion.

In addition, we made the change as advised (line 149, 342, 398). We use different font for the captions in this revised version.

Reviewer 3 Report

The work presented by  Jiasen He, et al.  at the Attention of the “Immuno” Editorial Board for publication examines in detail the clinical literature of CAR-T cell therapy in pediatric solid tumors by an interest point of view: the possibilities that oncolytic viruses may  promote anti-tumor responses through a dual mechanism of action that is dependent on selective tumor cell killing and the induction of systemic anti-tumor immunity.

While it has been demonstrated that oncolytic viruses could be usefully integrated into tumor immunotherapies, as they target multiple steps within the cancer–immunity cycle, their effects on cancer cells can facilitate CAR-T cells ACT by conditioning both the systemic immune system and local tumor microenvironment (TME) to better support T cell recruitment and effector function.

Those concepts have been well described by the authors in the flow-chart of the manuscript, allowing the reader to refer to specific preclinical studies or clinical trials, sometimes summarized with salient clinical and therapeutic information.  

Less attention has been dedicated to potential hurdles linked to utilization of OVs in clinical, possibly related to their effects possibly restricted by the tumor microenvironment, host-mounted anti-viral responses or pre-existing neutralizing antibodies, or side and toxic effects which I will really suggest to mention, maybe including a tab.

Other than that I believe that the originality of the manuscript, which is well presented by the authors, well written and fluent in his development, make this work suitable for publication with minor revision.  

Author Response

We appreciate the comments and suggestions from the reviewer.

As suggested by the reviewer, we added discussion on the potential hurdles to translate oncolytic virotherapy into clinic (lines 281-296). Since it’s not the focus of this review, we play more emphasis on the effect of the OVs relevant to combination with CAR T cell therapy.